# The CXCR6-CXCL16 axis mediates T cell control of polyomavirus infection in the kidney

Matthew D. Lauver[1☉], Zoe E. Katz[1☉], Havell Markus[2], Nicole M. Derosia[1], Ge Jin[1], Katelyn N. Ayers[1], Arrienne B. Butic[1], Kaitlyn Bushey[3], Catherine S. Abendroth[4], Dajiang J. Liu[2], Aron E. Lukacher[1]*

1 Department of Microbiology and Immunology, Penn State College of Medicine, Hershey, Pennsylvania, United States of America, 2 Department of Public Health Sciences, Penn State College of Medicine, Hershey, Pennsylvania, United States of America, 3 Bio X Cell, Inc., Lebanon, New Hampshire, United States of America, 4 Department of Pathology, Penn State Health Milton S. Hershey Medical Center, Hershey, Pennsylvania, United States of America

☉ These authors contributed equally to this work.
* ael17@psu.edu

## Abstract

BK polyomavirus (PyV) establishes lifelong asymptomatic infections in the reno-urinary system of most humans. BKPyV-associated nephropathy is the leading infectious cause of kidney allograft loss. Using mouse PyV, a natural murine pathogen that also persists in the kidney, we define a dominant chemokine receptor-chemokine axis that directs T cell infiltration of the kidney. We found that CXCR6 was required for CD4+ and CD8+ T cells to be recruited to and retained in the kidney, respectively. Absence of CXCR6 impaired virus control in the kidney. The soluble form of CXCL16 was increased in kidneys of infected mice and *in vivo* CXCL16 neutralization reduced numbers of virus-specific CD8+ T cells infiltrating the kidney. *In vivo* administration of IL-12 upregulated CXCR6 expression on virus-specific CD8+ T cells, improved T cell recruitment to the infected kidney, and reduced virus levels. Notably, T cells in kidney biopsies from PyV-associated nephropathy patients express CXCR6 and transcriptional analysis shows significant upregulation of *CXCR6* and *CXCL16*. These findings demonstrate the importance of the CXCR6-CXCL16 axis in regulating T cell responses in the kidney to PyV infection.

## Author summary

Polyomaviruses (PyV) establish lifelong silent infections in the kidneys of most people. In kidney transplant patients, however, BKPyV can resurge leading to dysfunction and loss of the renal allograft. BKPyV infection is controlled by the antiviral T cell response. How T cells home to the PyV-infected kidney to prevent viral reactivation is unknown. Using mouse PyV, a natural murine pathogen that persists in the kidney, we identified the CXCR6 chemokine receptor-CXCL16 chemokine axis as the key mechanism for recruitment and maintenance of kidney-infiltrating CD4+ and CD8+ T cells to keep virus infection in the kidney in check. We also found that administration of

**Data availability statement:** All data are available in the main text or the supplementary materials.

**Funding:** This work was supported by NIH grants R35NS127217 (AEL); 5T32GM102057 and F30GM151848 (HM); and R01ES036042, R01HG011035, and R01AI174108 (DJL). The funders had no role in the study design, data collection and analysis, decision to publish, or preparation of this manuscript.

**Competing interests:** The authors have declared that no competing interests exist.

IL-12 to infected mice upregulated expression of CXCR6 on virus-specific T cells and promoted their recruitment to the kidney. Bioinformatics analysis of bulk mRNA from kidney needle biopsies of transplant patients showed upregulated expression of *CXCR6* and *CXCL16* transcripts. Our findings define a specific chemokine receptor-chemokine axis required for T cell-mediated control of PyV infection in the kidney and elevate the potential for adoptive cellular immunotherapy to limit PyV resurgence in kidney transplant patients.

## Introduction

T cell surveillance is critical for controlling viral infections in non-lymphoid organs [1,2]. The protective capabilities of antiviral T cells to acute infections in barrier tissues (e.g., skin, respiratory and gastrointestinal mucosae) are well documented [3–5]. Less understood, however, is the entry and maintenance of T cells in virus-infected non-barrier tissues, such as the kidney. The kidney is the reservoir for persistence for a number of viruses that infect healthy humans and can cause severe disease upon dampened immune status [6]. Virus-specific T cells infiltrate and establish long-term residence in the kidney after acute viral infections [7,8]. The mechanisms motivating antiviral T cells to home to the kidney and prevent resurgence of persistent viral infections are incompletely understood.

Polyomaviruses (PyV) are ubiquitous human pathogens that cause lifelong persistent infections. Nearly all adults are seropositive for BKPyV, which resides within the epithelium of the reno-urinary tract [9,10]. Although asymptomatic in healthy individuals, BKPyV causes polyomavirus-associated nephropathy (PVAN) in kidney transplant (KTx) recipients [11,12]. The leading infectious cause of renal allograft loss, PVAN occurs in up to 14% of KTx recipients and results in graft loss in 50% of cases [12,13]. Because there are currently no FDA-approved antivirals for PVAN, treatment involves reduction of immunosuppression to improve cellular immune responses toBKPyV infection [14]. Enhancing immune responses increases the risk of allograft damage and rejection; thus, increasing T cell responses during PVAN requires a balance of reducing viral burden while preserving kidney function. Despite the importance of kidney-localized T cells in controlling persistent BKPyV infection and resolving PVAN, little is known about PyV-specific T cell responses in the kidney, in particular how these cells are recruited into and maintained in the kidney.

Particular chemokine receptors and their ligands control T cell responses to viral infections in different organs. CXCR3 and its ligands CXCL9, CXCL10, and CXCL11 are required for T cell responses in barrier tissues such as the skin and female reproductive tract [15,16]. CXCR3 has also been shown to control the accumulation of CD8+ T cells in the kidney following LCMV infection [8]. More recently, CXCR6 and its ligand CXCL16 have been implicated in the generation of memory T cells in the lung and brain in response to influenza and West Nile virus, respectively [17,18]. Whether these chemokine receptors or others contribute to the anti-PyV kidney T cell response remains to be elucidated.

Because polyomaviruses are species-specific, we have an incomplete picture of control of persistent BKPyV infection in the kidney in healthy individuals and its pathogenesis in transplant recipients. Our group has utilized mouse polyomavirus (MuPyV) to study PyV immunity and pathogenesis [19–22]. MuPyV shares many features with BKPyV, including infection of the kidney epithelium, asymptomatic persistent infection, and control by antiviral T cell responses [20]. In this study, we used MuPyV to investigate the chemokine receptor

responsible for mediating kidney T cell responses to PyV infection. We found that T cells infiltrating the kidney preferentially expressed CXCR6 compared to vascular T cells. Absence of CXCR6 led to reduced numbers of kidney-infiltrating T cells and elevated virus levels. MuPyV infection induced CXCR6 expression, which could be further increased by treatment with recombinant IL-12. In PVAN, kidney CD8+ T cells express CXCR6 and published RNA-seq datasets from PVAN kidney biopsies indicate significant increase in *CXCR6* and *CXCL16* transcripts. These data reveal that the CXCR6-CXCL16 axis regulates T cell responses in the kidney during PyV infection.

## Results

### CXCR6 is enriched on kidney-infiltrating T cells during infection

BKPyV and MuPyV are epitheliotropic viruses that replicate within the kidney nephron [20,23]. Virus-specific T cells must migrate into the kidney parenchyma to engage infected cells in renal tubular epithelium. To identify chemokine receptors expressed by kidney-infiltrating T cells, we performed intravascular (IV) labeling with FITC-CD45 mAb immediately prior to euthanasia of MuPyV-infected mice 30 days post infection (dpi). Kidney-infiltrating [i.e., FITC-CD45 mAb negative (IV-)] CD8+ T cells specific for the dominant MuPyV D^b-LT359 epitope had elevated CD69 expression relative to their IV+ counterparts, indicative of tissue retention (S1A Fig). To determine whether particular chemokine receptors are involved in recruitment of T cells to the MuPyV-infected kidney, we selected nine chemokine receptors reported to guide T cell migration to the inflamed kidney and to infection of other non-lymphoid organs [15,17,24,25]. We found higher levels of CXCR4, CXCR5, CCR5, and CCR6 on vascular CD8+ T cells than on infiltrating cells (Fig 1A). CXCR6, however, was the only receptor that displayed higher expression within the infiltrating CD8+ T cell population relative to the intravascular population. We also examined total activated CD4+ T cells, instead of virus-specific CD4+ T cells, because MHC II tetramers selectively bind to CD4+ T cells having high-affinity TCRs [22,26]. Similar to CD8+ T cells, CD69 was expressed at high levels on infiltrating CD4+ T cells (S1B Fig). Although CCR6 was elevated on intravascular CD4+ T cells, CXCR6 was increased on infiltrating cells as seen on the IV- CD8+ T cells (Fig 1B). To confirm that virus-specific CD4+ T cells expressed CXCR6, we stained cells with a combination of two tetramers for the dominant I-A^b epitopes in MuPyV (Fig 1C). CXCR6 was enriched on kidney-infiltrating MHC-II tetramer+ CD4+ T cells compared to intravascular cells, indicating that CXCR6 is preferentially expressed on both virus-specific CD8+ and CD4+ T cells that infiltrate the kidney (Fig 1D). Tracking kidney T cell CXCR6 expression over time, we observed consistent expression of CXCR6 in CD4+ T cells at 8, 15, and 30 dpi. For CD8+ T cells, expression increased from 8 to 15 dpi and was then maintained to 30 dpi (Fig 1E). Using a CXCR6-GFP reporter mouse, we confirmed CXCR6 expression on CD8+ T cells within the kidney (Fig 1F).

As CXCL16 is the sole ligand for CXCR6, we next examined expression of CXCL16 in the kidney. CXCL16 mRNA was unchanged in the kidney following infection and CXCL16 protein expression was detectable throughout the kidney epithelium both in sham- and MuPyV-infected mice by immunofluorescence (Fig 2A and 2B). This constitutive, basal expression of CXCL16 in kidney epithelial cells is consistent with previous studies of CXCL16 expression during kidney injury [27,28]. CXCL16 is produced as a transmembrane protein (mCXCL16) whose external domain is cleaved to produce the soluble form of CXCL16 (sCXCL16). Examining CXCL16 protein expression in kidney homogenates, we observed expression of several forms of the uncleaved mCXCL16 in sham and acutely infected mice: two bands at approximately 65 kDa due to differentially glycosylated mCXCL16 [29] and a

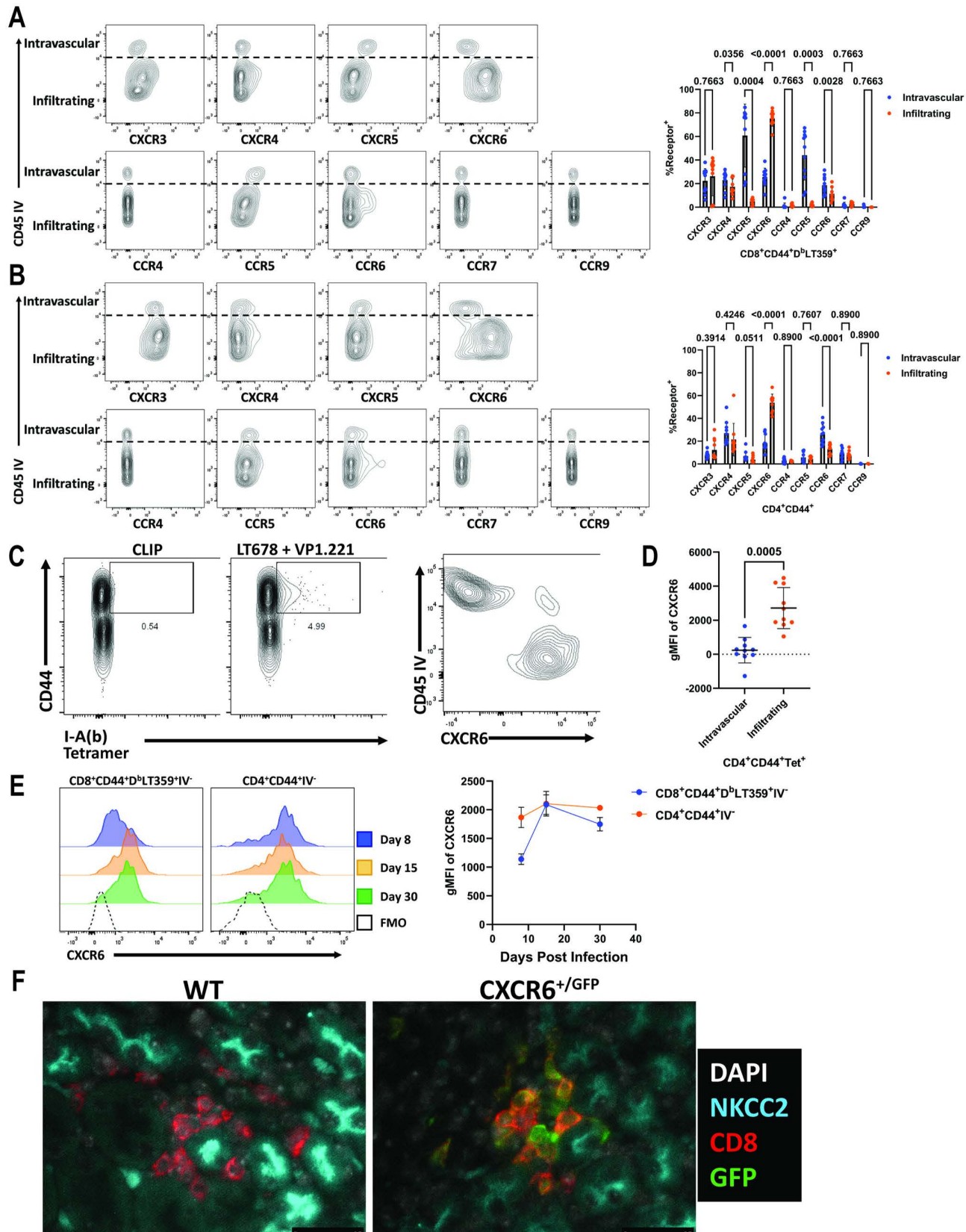

**Fig 1. CXCR6 is expressed on kidney-infiltrating T cells.** (A) Representative flow plots of chemokine receptor expression versus intravenous labeling on CD8[+] CD44[+] D[b]-LT359 tetramer[+] T cells (left). Frequency of receptor[+] intravascular and infiltrating CD8[+] CD44[+] D[b]-LT359 tetramer[+] T cells

(right). Data are from three independent experiments (n = 9-10). (B) Representative flow plots of chemokine receptor expression versus intravenous labeling on CD4+ CD44+ T cells (left). Frequency of receptor+ intravascular and infiltrating CD4+ CD44+ T cells (right). Data are from three independent experiments (n = 9-10). (C) Control (CLIP) and MuPyV-specific I-A^b tetramer staining on kidney CD4+ CD44+ T cells 30 dpi. (D) CXCR6 expression on intravascular and infiltrating CD4+ CD44+ tetramer+ T cells 30 dpi. (E) CXCR6 expression over time on infiltrating CD8+ CD44+ D^b-LT359 tetramer+ and CD4+ CD44+ T cells. Data are from two independent experiments (n = 10, C and D) and of three independent experiments (n = 3-4, A-B, E). (F) Staining for GFP in the kidneys of WT and CXCR6+/GFP mice. Data were analyzed by multiple paired *t* tests (A and B) and by paired *t* tests (D).

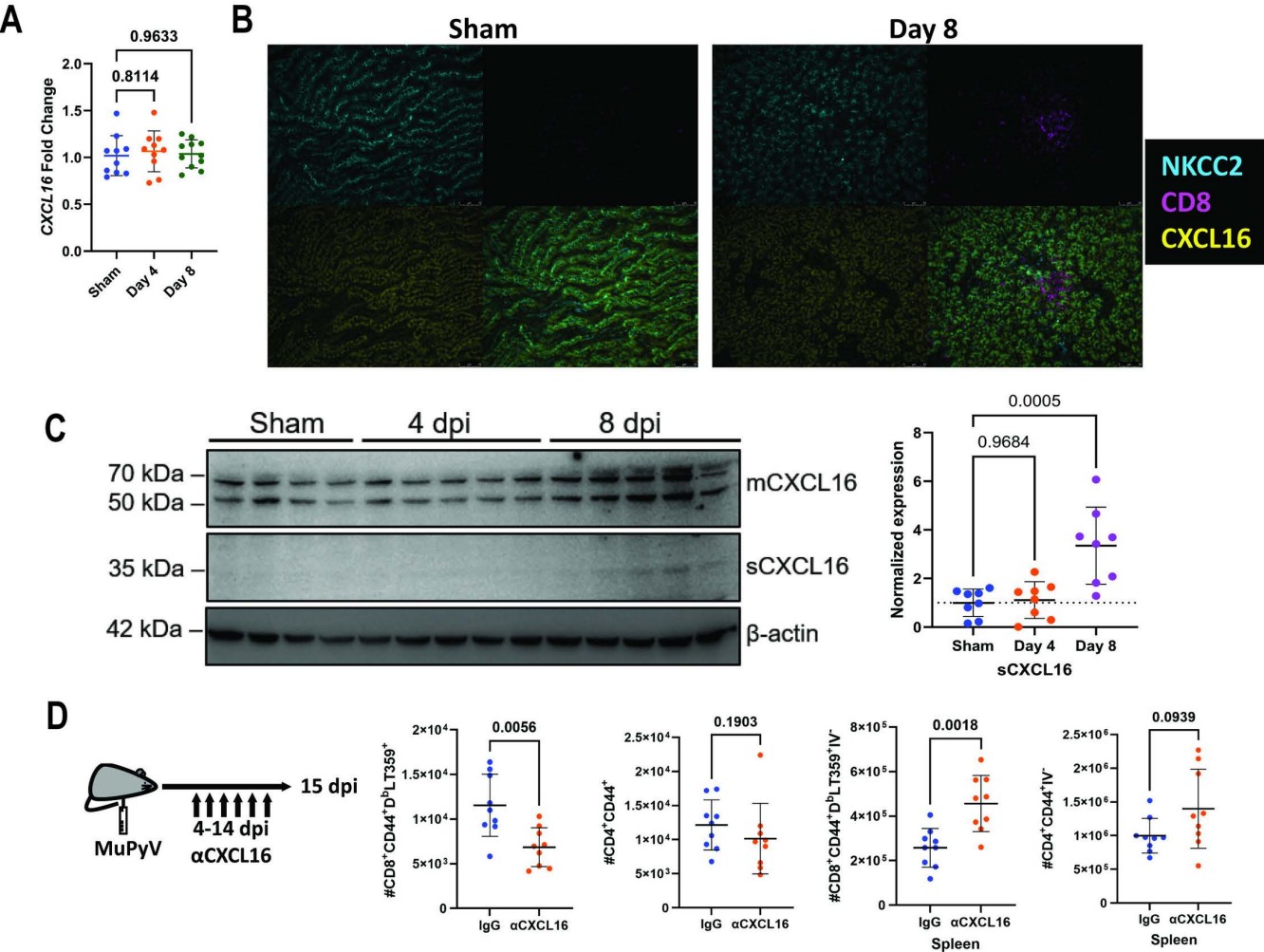

**Fig 2. CXCL16 is upregulated in kidneys during MuPyV infection and is required for T cell kidney infiltration.** (A) CXCL16 mRNA expression in the kidney during MuPyV infection. Expression is shown as fold change relative to sham infected samples. Data are from three independent experiments (n = 10-11). (B) Expression of NKCC2 [marks the ascending loop of Henle [56]], CD8, and CXCL16 in epithelia in sham-infected and 8 dpi kidneys; bottom right photomicrographs are merged images. Representative of two independent experiments. (C) CXCL16 expression in sham-infected, 4 dpi, and 8 dpi kidney lysates (left). Western blot image is representative of two independent experiments with each lane indicating protein lysate from kidneys of individual mice. Protein band intensity quantification for sCXCL16 was normalized to β-actin and analyzed by ImageLab and normalized to the loading control (right). Data are combined from two independent experiments (n = 3-5). (D) Experimental design of *in vivo* CXCL16 mAb administration. Mice were administered 250 μg of CXCL16 mAb or control rat IgG every two days from days 4-14 post infection and euthanized at 15 dpi. Numbers of CD45 mAb i.v.-negative, CD8+ CD44+ D^b-LT359 tetramer+ T cells and CD4+ CD44+ T cells in kidneys and spleens of infected mice given anti-CXCL16 or control rat IgG. Data were analyzed by one-way ANOVA (A and C) and by multiple Mann-Whitney tests (D).

50 kDa non-glycosylated form [30]. The 35 kDa sCXCL16 was elevated at 8 dpi in kidneys of three of five mice as shown by the increased band intensities normalized against a β-actin loading control. These data indicate an increase in CXCL16-mediated chemoattraction in the MuPyV-infected kidney (Fig 2C).

To confirm that CXCL16 was involved T cell kidney infiltration, mice were given a neutralizing CXCL16 mAb on alternate days over a two-week period starting 4 dpi. As shown in Fig 2D, numbers of infiltrating (FITC-CD45 mAb IV⁻) D^b-LT359 tetramer⁺ CD8⁺ T cells in the kidney were significantly lower; kidney-infiltrating CD4⁺ T cell numbers were lower as well, but the difference with infected control mice did not reach statistical significance. Conversely, more virus-specific CD8⁺ T cells were present in the spleen in CXCL16 mAb-treated mice than in control mice receiving rat IgG, and CD4⁺ T cell numbers were higher but did not attain statistical significance. Together, these data demonstrate that T cell infiltration into the kidneys of MuPyV-infected mice is dependent on the CXCR6-CXCL16 axis.

## Absence of CXCR6 impairs kidney T cell responses and virus control

The preferential expression of CXCR6 on kidney-infiltrating T cells and CXCL16 expression in the kidney led us to ask whether CXCR6 was required for T cell migration into the parenchyma. To assess the impact of CXCR6 on T cell trafficking, we compared T cells in the kidneys of WT and CXCR6^{−/−} mice 8, 15, and 30 dpi. Infiltrating CXCR6^{−/−} kidney T cells had no detectable CXCR6 compared to intravascular T cells but elevated CXCR6 expression was seen on kidney-infiltrating T cells in WT mice relative to intravascular cells (S2A Fig). CXCR6^{−/−} mice had equivalent numbers of splenic virus-specific CD8⁺ T cells and activated CD4⁺ T cells to WT mice (S2B–S2C). In the kidney, however, CXCR6^{−/−} mice had similar infiltrating CD8⁺ T cell numbers at 8 dpi, but decreased numbers at 15 and 30 dpi (Fig 3A). In contrast, infiltrating CD4⁺ T cells were reduced in the CXCR6^{−/−} mice from day 8 dpi onward (Fig 3B).

Reduced T cell numbers in the CXCR6^{−/−} kidneys led us to ask whether the absence of CXCR6 altered virus control. Viral DNA levels were unchanged in the spleen over the course of infection, indicating that loss of CXCR6 did not result in a global defect in virus control (S2D Fig). In marked contrast, virus levels were higher in CXCR6⁻ mice, demonstrating the requirement of CXCR6 for virus control in the kidney (Fig 3C). No differences in virus levels were observed in either the spleen or kidney at 4 dpi between WT and CXCR6^{−/−} mice, indicating that impaired virus control occurs concomitant with the appearance of T cells in the kidney (S2E and S2F Fig). To determine if the loss of CXCR6 affected virus levels in other non-lymphoid organs, we examined virus levels in the liver, lung, and salivary gland 30 dpi. We observed a modest elevation in virus levels in the liver, the organ with the lowest virus levels of those examined, and no effect on virus levels in the lung and salivary gland, indicating that CXCR6⁺ T cells predominantly control virus levels in the kidney (S2G Fig).

The antibody response to PyV is crucial for controlling infection in the kidney [20,31]. To determine whether loss of CXCR6 impaired virus antibody production, we measured serum viral VP1 capsid antibody levels throughout infection and found no difference between WT and CXCR6^{−/−} mice, excluding a defect in humoral immunity as the cause of elevated kidney virus levels (S2H Fig). These data demonstrate that CXCR6 is required for proper T cell accumulation and localization in the kidney to mediate virus control.

The impact of CXCR6 on CD8⁺ T cell maintenance in the kidney next led us to ask whether this requirement was cell-intrinsic. To test this, we crossed CXCR6^{−/−} to TCR-I TCR transgenic mice, which recognize the D^b-LT206 epitope expressed in a recombinant MuPyV, designated A2.LT206. We transferred a 1:1 mix of WT and CXCR6^{−/−} TCR-I CD8⁺ T cells

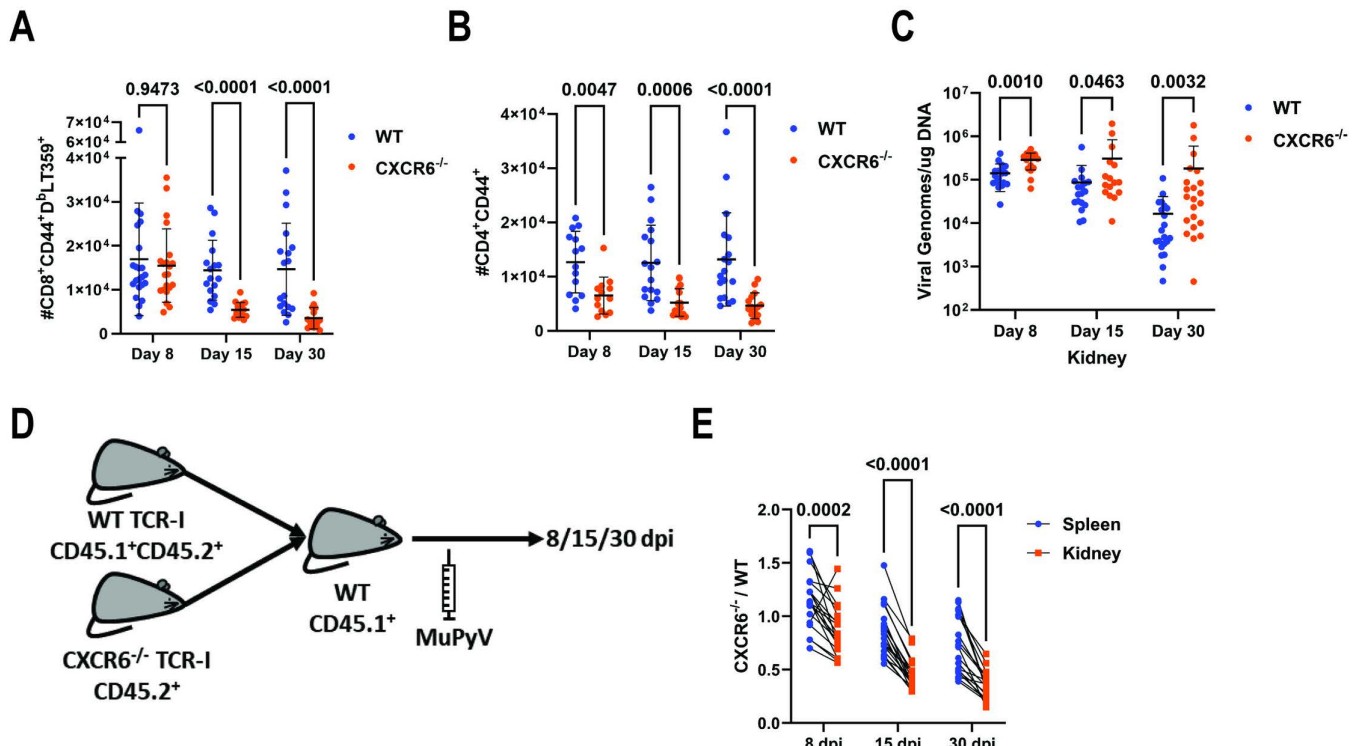

**Fig 3. CXCR6 is required for T cell maintenance and virus control.** (A) Number of CD8+ CD44+ Db-LT359 tetramer+ T cells infiltrating the kidney parenchyma of WT and CXCR6−/− mice 8, 15, and 30 dpi. Data are from 4-5 independent experiments (n = 15-22). (B) Number of CD4+ CD44+ T cells infiltrating the kidney parenchyma of WT and CXCR6−/− mice 8, 15, and 30 dpi. Data are from 3-4 independent experiments (n = 13-18). (C) Number of Viral DNA genomes in the kidney of WT and CXCR6−/− mice 8, 15, and 30 dpi. Data are from 4-5 independent experiments (n = 12-22). (D) Experimental design for mixed adoptive transfer of WT and CXCR6−/− TCR-I cells. CD45.1+ recipients received 1 x 10⁴ total TCR-I cells (5,000 WT:5,000 CXCR6−/−) one day prior to infection with MuPyV. (E) Ratio of total CXCR6−/− TCR-I cells compared to total WT TCR-I cells in the kidney and spleen at 8, 15, and 30 dpi. Data are from four independent experiments (n = 19). Data were analyzed by multiple Mann-Whitney tests (A-C) or multiple paired *t* tests (E).

into WT recipient mice and infected the recipients with A2.LT206 (Fig 3D). Examining the frequency of CXCR6−/− TCRI cells in the spleens and kidneys of the recipients at 8, 15, and 30 dpi, we found a reduced frequency of CXCR6−/− cells in the kidney relative to the spleen at all three timepoints, with the frequency decreasing over time (Fig 3E). These data demonstrate a CD8+ T cell-intrinsic function of CXCR6 in maintaining T cell responses in the kidney during MuPyV infection.

## Lack of CXCR6 leads to failed retention of CD8+ T cells

The reduction in kidney T cell numbers in the CXCR6−/− mice raised the possibility of changes in cell proliferation or apoptosis bearing responsibility for this reduction. To address this, we examined Ki67 expression in the kidney T cells of WT and CXCR6−/− mice. Despite having fewer cells, kidney CXCR6−/− CD4+ and CD8+ T cells had a higher frequency of Ki67+ cells than in infected WT kidneys (Fig 4A and 4B). This increase in Ki67+ cells was cell-intrinsic, as CXCR6−/− TCR-I CD8+ T cells transferred 1:1 with WT cells also displayed elevated Ki67 at 15 and 30 dpi in the kidney but not in the spleen (Figs 4C and S3A). To determine if this increased proliferation was balanced by increased apoptosis in the CXCR6−/− T cells, kidney T cells at 15 dpi were stained with Annexin V or incubated with FLICA reagent to measure apoptosis and active caspase 3/7 activity, respectively.

CXCR6$^{-/-}$ kidney T cells displayed similar levels of Annexin V binding and caspase 3/7 activity as WT T cells, indicating that cell death by apoptosis was not responsible for loss of kidney CXCR6$^{-/-}$ T cells (Figs 4D and S3B). The reduced cell numbers at 15 dpi were also not the result of increased T cell death in the kidney at 8 dpi, as no increase in Annexin V staining was observed, indicating there is not an equivalent increase in apoptosis to counterbalance the increase in proliferation (S3C Fig).

Because the kidney serves as a reservoir for persistent MuPyV infection, we considered the possibility that the Ki67$^+$ cells in the CXCR6$^{-/-}$ mice may have recently migrated into the

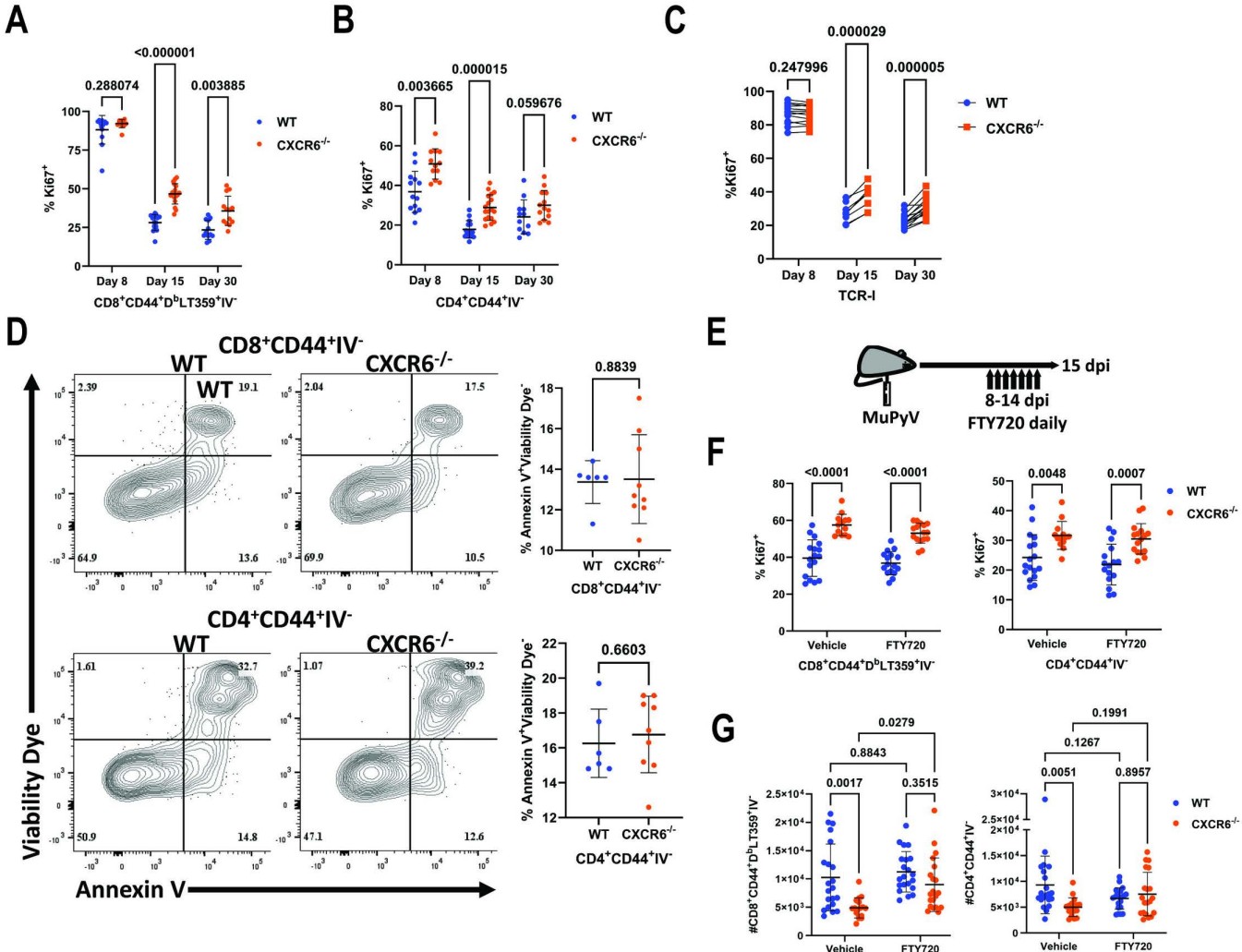

**Fig 4. CXCR6 is required for retention of kidney-infiltrating CD8$^+$ T cells.** (A) Frequency of kidney-infiltrating CD8$^+$ CD44$^+$ D$^b$LT359 tetramer$^+$ T cells that are Ki67$^+$ in WT and CXCR6$^{-/-}$ mice at 8, 15, and 30 dpi. Data are from 3-4 independent experiments (n = 12-18). (B) Frequency of kidney-infiltrating CD4$^+$CD44$^+$ T cells that are Ki67$^+$ in WT and CXCR6$^{-/-}$ mice at 8, 15, and 30 dpi. Data are from 3-4 independent experiments (n = 12-18). (C) Frequency of Ki67$^+$ WT and CXCR6$^{-/-}$ kidney-infiltrating TCR-I cells 8, 15, and 30 dpi. Data are from 2-3 independent experiments (n = 9-15). (D) Kidney CD8$^+$ and CD4$^+$ T cells stained with viability dye and Annexin V to identify apoptotic (Annexin V$^+$ viability dye$^-$) cells in WT and CXCR6$^{-/-}$ mice 15 dpi. Data are from two independent experiments (n = 6-9). (E) Experimental design for FTY720 treatment. Mice received 1 mg/kg of FTY720 or vehicle control daily from 8-14 dpi and were euthanized at 15 dpi (top). (F) Frequency of kidney-infiltrating CD8$^+$ CD44$^+$ D$^b$-LT359 tetramer$^+$ and CD4$^+$ CD44$^+$ T cells that are Ki67$^+$ in WT and CXCR6$^{-/-}$ mice with vehicle or FTY720 treatment. Data are from three independent experiments (n = 10-13). (G) Numbers of kidney-infiltrating CD8$^+$ CD44$^+$ D$^b$-LT359 tetramer$^+$ and CD4$^+$ CD44$^+$ T cells in vehicle and FTY720-treated mice. Data were analyzed by multiple Mann-Whitney tests (A-B), multiple paired $t$ tests (C), unpaired $t$ tests (D), multiple unpaired $t$ tests (E), and two-way ANOVA (F).

kidney. To test this, we treated WT and CXCR6⁻/⁻ mice with FTY720 daily days 8-14 post-infection to prevent migration of new T cells into the kidney (Fig 4E). At 15 dpi, FTY720-treated mice had significantly reduced levels of circulating CD4⁺ and CD8⁺ T cells (S3D Fig); however, kidney T cells in the CXCR6⁻/⁻ mice retained elevated Ki67 expression, confirming the T cells are proliferating *in situ* (Figs 4F and S3E). FTY720-treatment depressed virus-specific CD8⁺ and CD4⁺ T cell responses in the spleen (S3F Fig). In the kidney however, FTY720-treated CXCR6⁻/⁻ mice had increased virus-specific CD8⁺, but not CD4⁺, T cell numbers compared to vehicle-treated CXCR6⁻/⁻ mice (Fig 4G). This indicates that the decrease in kidney CD8⁺ T cells in CXCR6⁻/⁻ mice is the result of a failure to retain these cells after entry into the kidney, as evidenced by the normal cell numbers at 8 dpi followed by decreased numbers at 15 and 30 dpi (Fig 3A). In contrast, FTY720 treatment did not rescue CD4⁺ T cell numbers, as these cells display a defect in recruitment with reduced numbers apparent at 8 dpi (Fig 3B).

## IL-12 promotes CXCR6 expression

The function of CXCR6 in kidney T cell maintenance and virus control led us to investigate how CXCR6 expression was regulated in the context of MuPyV infection. Based on the constitutive basal expression of CXCL16 by the kidney epithelia, we examined whether MuPyV infection increased CD8⁺ T cell expression of CXCR6 in the spleen. MuPyV-specific CD8⁺ T cells in the spleen expressed higher levels of CXCR6 than activated CD8⁺ T cells in uninfected mice (Fig 5A). CXCR6 expression was higher on Dᵇ-LT359 tetramer⁺ CD8⁺ T cells than activated, tetramer-negative CD8⁺ T cells (S4A Fig).

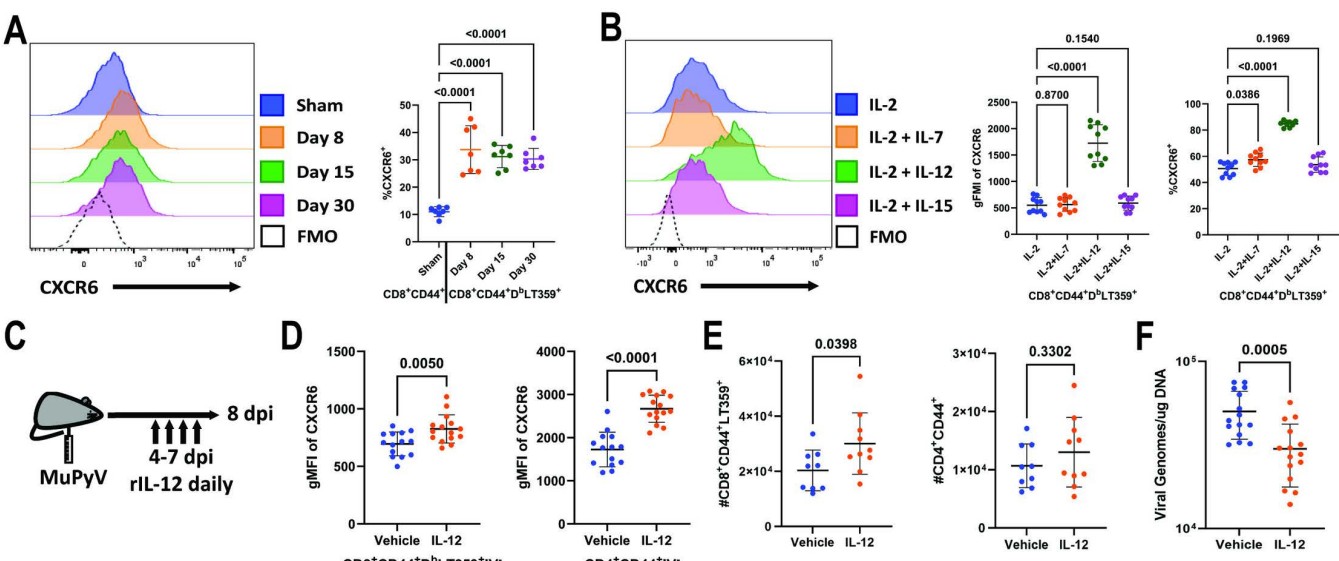

**Fig 5. IL-12 induces CXCR6 on PyV-specific T cells.** (A) CXCR6 expression on FITC-CD45 mAb i.v. negative, splenic CD8⁺ CD44⁺ (sham) or CD8⁺ CD44⁺ Dᵇ-LT359 tetramer⁺ (8, 15, 30 dpi) T cells. Data are from two independent experiments (n = 7). (B) CXCR6 expression on splenic CD8⁺ CD44⁺ Dᵇ-LT359 tetramer⁺ T cells isolated from WT mice 8 dpi and cultured overnight with the indicated cytokines. Data are from two independent experiments (n = 10). (C) Experimental setup for IL-12 administration. (D) CXCR6 expression on kidney-infiltrating CD8⁺CD44⁺ DᵇLT359 tetramer⁺ T cells (left) on kidney-infiltrating CD4⁺CD44⁺ T cells (right). Data are from three independent experiments (n = 14-15). (E) Numbers of CD8⁺CD44⁺ Dᵇ-LT359 tetramer⁺ (left) and CD4⁺ CD44⁺ T cells (right) in kidney after vehicle or IL-12 administration. Data are from two independent experiments (n = 9). (F) Number of viral DNA genomes in kidney after vehicle or IL-12 administration. Data are from three independent experiments (n = 14-15). Data were analyzed by one-way ANOVA (A), repeated measures one-way ANOVA (B), or unpaired *t* tests (D-F).

Previous work has implicated IL-12 and IL-15 in the induction of CXCR6 on T cells and NK cells [32,33]. To determine whether either of these cytokines influence CXCR6 expression on virus-specific T cells, splenocytes from mice 8 dpi were cultured overnight in the presence of IL-2 +/- IL-12 or IL-15. Addition of IL-15 did not affect CXCR6 expression, whereas IL-12 induced CXCR6 on virus-specific CD8+ T cells (Fig 5B). Given this *ex vivo* induction of CXCR6 by IL-12, we next asked whether IL-12 would increase CD8+ T cell expression of CXCR6 *in vivo*. Beginning 4 dpi, mice were given daily injections of IL-12 and examined for CXCR6 expression 8 dpi (Fig 5C). IL-12 treatment increased CXCR6 expression on virus-specific CD8+ T cells in the kidney, but not in the spleen (Figs 5D and S4B). For antigen-experienced CD4+ T cells, IL-12 treatment increased CXCR6 expression in both the spleen and kidney (Figs 5D and S4B). Concomitantly, numbers of antiviral CD8+ T cells in the kidney, but not the spleen, were also higher in mice given IL-12 compared to vehicle control; no differences in CD4+ T cell numbers were observed in either the kidney or the spleen after

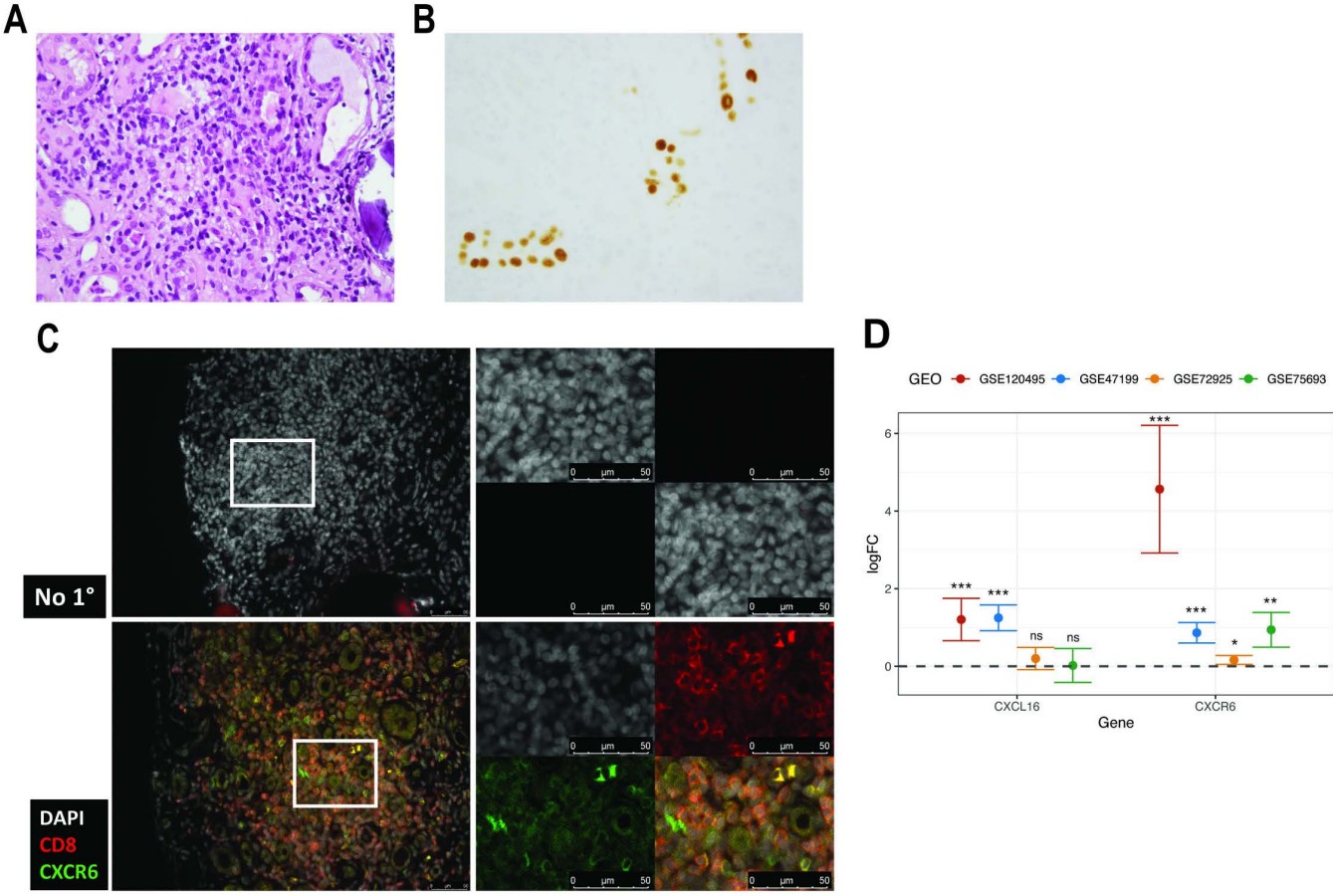

**Fig 6. CXCR6 is expressed by CD8+ T cells during PVAN.** (A) Dense lymphoplasmacytic infiltrate surrounding and infiltrating cortical tubules (H&E, x400). (B) Immunohistochemical staining for Large T antigen in nuclei of tubular epithelium (x250). (C) Expression of CXCR6 on infiltrating CD8+ cells. Photomicrographs (right) are enlarged images in the white squares (left); bottom right are merged images. No 1°, no primary antibody. (D) Log-fold change of CXCL16 and CXCR6 across four independent studies, each comparing KTx biopsies from patients with PVAN and stable graft function. The error bars indicate the 95% confidence interval of log-fold change, with horizontal dashed grey line indicating log-fold change = 0, or no difference. GSE120495 is an RNA-seq study, while remaining studies are microarray based. Given the heterogeneity of the data, a Cauchy combination method with uniform weights was employed to integrate the p-values across these studies. The combined p-values for CXCL16 = 0.000337 and CXCR6 = 2.72x10$^{-05}$. ns, non-significant at adjusted p-value > 0.05; *, statistical significance at adjusted p-value ≤ 0.05; **, statistical significance at adjusted p-value ≤ 0.01; ***, statistical significance at adjusted p-value ≤ 0.001.

IL-12 treatment (Figs 5E and S4C). After IL-12 treatment, virus levels in the spleen were not affected, however, they were decreased in the kidney compared to vehicle administration (Figs 5F and S4D). Together, these data demonstrate that IL-12 upregulates expression of CXCR6 both *in vitro* and in MuPyV-infected mice.

## Human kidney CD8+ T cells express CXCR6 during PVAN

The importance of kidney T cell CXCR6 expression during MuPyV infection led us to ask whether CXCR6 was expressed by human T cells during PVAN. We obtained a renal needle biopsy from a KTx recipient with a diffuse interstitial infiltrate of lymphocytes and plasma cells, associated with severe tubulitis (Fig 6A). Anti-Large T antigen immunohistochemical staining decorated numerous tubular epithelial cell nuclei, with Large T antigen+ epithelial cells adjacent to each other within the same tubules (Fig 6B). No glomerulitis or peritubular capillaritis was appreciated. Moderate interstitial fibrosis/tubular atrophy affected approximately 50% of the cortex and there was mild vascular intimal fibrosis. Staining for CD8 and CXCR6 revealed numerous CD8+ cells within the lymphocytic aggregates and widespread CXCR6 expression on lymphoid cells, including on the CD8+ cells (Fig 6C). This indicates that CXCR6 is expressed on human CD8+ T cells infiltrating the kidney during PVAN.

To validate involvement of CXCR6 and CXCL16 during PVAN, we utilized previously published expression data from kidney biopsy samples of patients who had undergone kidney transplantation and subsequently developed PVAN and samples from patients with stable graft function. We identified four relevant previously published studies, of which one was RNA-seq based and three were microarray-based. *CXCL16* was significantly upregulated in kidney samples of PVAN patients as compared to patients with stable graft function in two of the four studies after false discovery rate (FDR) correction (Fig 6D and Table 1). Similarly, *CXCR6* was significantly upregulated in all four studies after FDR correction compared to patients with stable graft function. To assess the overall significance of these findings, we performed a meta-analysis using the Cauchy combination method with uniform weights to integrate the p-values across studies [34]. The meta-analysis confirmed the significant upregulation of both *CXCL16* (combined p-value = 0.000337) and *CXCR6* (combined p-value = 2.72x10$^{-05}$) in the kidney samples of PVAN patients (Table 1). Overall, these results support CXCR6 signaling in mediating T cell responses in the PyV-infected kidney.

**Table 1. Summary of differential gene expression of CXCL16 and CXCR6 in four studies of KTx biopsies with PVAN. Because of the heterogeneity of the data (i.e., one RNA-seq, three microarray), a Cauchy combination method with uniform weights was employed to integrate the p-values across these studies. P-values significant after false discover rate (FDR) correction are bolded and italicized in the Adj. P-value column. Cauchy combined p-values < 0.05 are bolded and italicized.**

| Study | Type | Case | Controls | Gene Symbol | P-Value | Adj. P-Value | T-statistic | log Fold Change | Cauchy Combined P-Value |
|---|---|---|---|---|---|---|---|---|---|
| GSE47199 | Array | 3 | 11 | *CXCL16* | 2.21755E-06 | *9.77602E-05* | 7.351291 | 1.2475493 | *0.000337503* |
| GSE72925 | Array | 10 | 68 | | 0.175292537 | 0.31964674 | 1.367734 | 0.2002145 | |
| GSE75693 | Array | 15 | 30 | | 0.927611847 | 0.958121013 | 0.09134005 | 0.02031828 | |
| GSE120495 | RNA-seq | 5 | 5 | | 1.56102E-05 | *0.00060813* | 4.319897 | 1.203815 | |
| GSE47199 | Array | 3 | 11 | *CXCR6* | 1.04958E-05 | *0.000330863* | 6.437358 | 0.8630277 | *2.72E-05* |
| GSE72925 | Array | 10 | 68 | | 0.007450893 | *0.0365479* | 2.747155 | 0.5892737 | |
| GSE75693 | Array | 15 | 30 | | 0.000149281 | *0.003696534* | 4.12765615 | 0.93965812 | |
| GSE120495 | RNA-seq | 5 | 5 | | 5.50963E-08 | *6.94882E-06* | 5.434027 | 4.562403 | |

## Discussion

In this study, we identified CXCR6 as a key mediator of T cell recruitment to the kidney following MuPyV infection. CXCR6 was enriched on kidney-infiltrating T cells and its sole ligand, CXCL16, was expressed in the kidney with its soluble form increased during acute infection. Absence of CXCR6 resulted in reduced recruitment of CD4+ T cells and impaired maintenance of CD8+ T cells in the kidney, which coincided with elevated virus levels specifically within the kidney. *In vivo* neutralization with a CXCL16 mAb resulted in lower numbers of virus-specific CD8+ T cell kidney infiltrates. The reduced kidney T cell response in CXCR6−/− mice was not due to increased cell death or decreased proliferation, but rather a failure of cell retention in the kidney, which could be ameliorated by blocking T cell emigration with FTY720. CXCR6 expression on PyV-specific T cells was induced by IL-12. IL-12 increased CXCR6 expression *in vitro* and *in vivo*, improved T cell recruitment to the infected kidney, and reduced virus levels. Importantly, CD8+ T cells express CXCR6 in a PVAN kidney and bulk RNAseq datasets from PVAN patients confirm upregulated expression of CXCR6 and CXCL16. Together, these findings not only provide mechanistic insight into the requirements for effective antiviral T cell control of PyV infection in the kidney, but also point toward a strategy to bolster virus-specific adaptive immunity and prevent virus resurgence in patients at risk for diseases caused by BKPyV and JCPyV that persist lifelong in the kidneys of most humans.

We and others have found that CXCL16 is basally expressed on kidney epithelial cells (Fig 2A and 2B) in the distal tubule and connecting tubule [35]. The tropism of the virus may contribute to the specific chemokine receptor-chemokine axis that the antiviral T cells follow after infection. MuPyV replicates in renal tubule epithelial cells with increased expression of the soluble form of CXCL16 [Fig 2C; [19,36]]. The transmembrane form of CXCL16 is cleaved by the metalloproteases ADAM10 and ADAM17 to yield soluble CXCL16 [35,37]. PyV infection may increase the expression and/or activity of these enzymes in the infected kidney epithelial cells. The CXCR6-CXCL16 axis is also involved in recruitment of T cells to the lungs and livers of mice infected with influenza virus and malaria, respectively [17,38]. An interesting possibility is that ADAM10 and ADAM17 direct trafficking of CXCR6+ T cells to different tissues as a consequence of their increased activity upon infection of parenchymal cells by particular pathogens.

Using the Armstrong strain of LCMV, Ma et al. reported CXCR3 expression on infiltrating kidney CD8+ T cells and a requirement for CXCR3 for T cell extravasation into the kidney [8]. We observed limited CXCR3 expression on kidney T cells, and no preferential expression on infiltrating cells relative to those in the vasculature (Fig 1A and 1B). This difference may indicate that CXCR6 is needed to maintain T cells in the kidney in the presence of persistent infection and prolonged inflammation. Recent work has identified CXCL16-CXCR6 signaling as the mediator of group 3 innate lymphoid cell accumulation in the kidney during renal fibrosis and chronic kidney disease, supporting a function of CXCR6 in kidney immune responses during persistent/chronic inflammation [39].

Absence of CXCR6 resulted in normal CD8+ T cell recruitment at day 8, but a failure to maintain these cells as numbers declined at day 15 and 30 (Fig 3A). In contrast, CD4+ T cells were reduced in CXCR6−/− mice from day 8 onward, indicating an earlier defect in recruitment (Fig 3B). Treatment of mice with FTY720 starting 8 dpi increased CD8+ T cell numbers at 15 dpi, implicating a failure of cell retention in the kidney as the explanation for declining cell numbers (Fig 4G). FTY720 treatment has previously been shown to block T cell migration from non-lymphoid tissues into afferent lymphatics [40]. Our data suggests that in the kidney CXCR6 signaling counteracts sphingosine 1-phosphate receptor signaling, which pulls CD8+ T cells out of the kidney. FTY720 treatment did not rescue

CD4[+] T cells in the kidney, as a defect in cell recruitment is already evident by 8 dpi when treatment began.

Our findings that exogenous IL-12 induced CXCR6 expression on T cells *in vitro* and *in vivo* are consistent with previously published data showing IL-12 treatment during *in vitro* T cell activation promotes CXCR6 expression [33]. In lymphocytes, IL-12 signaling triggers the phosphorylation and activation of STAT4 [41]. Although a direct connection between STAT4 activation and CXCR6 expression has not been shown, loss of STAT4 in ILC1 and NK cells leads to a reduction in CXCR6 expression [42]. Our data support a model in which IL-12 signaling through STAT4 leads to an upregulation of CXCR6 to enhance kidney homing and retention. IL-12 treatment has generated interest as an antitumor therapy, but failed to gain traction due to the toxicity of systemic IL-12 administration [43–45]. The more recent advent of CAR-T cell therapies has renewed interest in localized IL-12 injections or adoptive cell therapies with IL-12 treatment [46–48]. Our data suggests that conditioning antiviral T cells with IL-12 prior to transfer could increase CXCR6-mediated migration to the kidney and improve virus control in KTx patients with resurgent BKPyV replication.

HLA-matched, virus-specific T cell therapy is actively being pursued to treat persistent viral infections, including BKPyV and JCPyV [49,50]. Managing PVAN, however, is challenging due to the need to balance improving antiviral T cell responses with avoiding anti-donor T cell responses. Notably, we showed that MuPyV infection in recipients of MHC class I and class II-mismatched kidneys increased the magnitude of the anti-donor T cell response [51]. Interrupting the CXCR6-CXCL16 axis by systemic administration of neutralizing CXCL16 mAb could be an efficacious intervention to reduce anti-donor T cell infiltrates in kidney allografts. To enhance the efficacy of adoptive T cell transfers in a model of pancreatic cancer, Lesch et al. demonstrated that retroviral transduction of *CXCR6* in antigen-specific T cells promoted their migration and binding to CXCL16-expressing tumor cells, which led to an improved anti-tumor response [52]. Similarly, enforced expression of CXCR6 in BKPyV-specific T cells could facilitate their infiltration into renal allografts with PVAN. Thus, this work has potentially valuable clinical implications for ameliorating PVAN.

## Materials and methods

### Ethics statement

All experiments conducted on mice were in accordance with approved protocols. Mouse studies and experimental protocols were approved by the Penn State College of Medicine Institutional Animal Care and Use Committee (Protocol no. PRAMS201447619).

### Mice

C57BL/6J ("WT", RRID:IMSR_JAX:000664), B6.129P2-Cxcr6[tm1Litt]/J ("CXCR6[−/−]", RRID:IMSR_JAX:005693), C57BL/6J-Ptprc[em6Lutzy]/J ("CD45.1", RRID:IMSR_JAX:033076) mice were purchased from Jackson Laboratories. TCR-I mice expressing a T cell receptor specific for the SV40 large T antigen epitope at amino acids 206-215 and the recombinant MuPyV carrying this epitope have been previously described [53,54]. Mice were housed and bred under specific pathogen-free conditions at the Penn State College of Medicine. Male and female mice 6-12 w of age were used for experiments.

### Virus infections and treatments

All experiments were performed using the A2 strain of MuPyV, except as indicated for A2.LT206, where the D[b]-LT359 epitope in MuPyV was changed by site-directed mutagenesis to the corresponding D[b]-LT206 epitope in SV40 Large T antigen recognized by

TCR-transgenic TCR-I CD8$^+$ T cells [54]. Mice were infected via the hind footpad with 1x10$^6$ PFU of MuPyV. For IL-12 treatments, mice received 250 ng of recombinant mouse IL-12 (Biolegend) daily i.p. from days 4-7 post infection or PBS vehicle control. For FTY720 treatment, mice received 1 mg/kg FTY720 (Sigma) daily i.p. from days 8-14 post infection or PBS vehicle control. For CXCL16 neutralization, mice were infected with MuPyV and received 250 μg of anti-mouse CXCL16 mAb (clone 12-81, BioXCell) i.p. on alternate days from days 4-14 post infection or control rat IgG (Jackson ImmunoResearch Laboratories).

## Cell isolation and flow cytometry

For flow cytometry experiments, circulating cells were i.v. labeled by retro-orbitally injecting mice with 3 μg of FITC-conjugated anti-CD45 (Biolegend, clone 30-F11) 3 m prior to euthanasia. For isolation of lymphocytes, kidneys were mechanically disaggregated and treated with DNAase (3 mg/100 mL) and Collagenase I (37 mg/100 mL) for 30 m. The tissue was then passed through 70 μm strainer and centrifuged on a 2-step (44%/66%) percoll gradient. MHC-II tetramer staining of CD4$^+$ T cells was performed with a combination of two PE-conjugated I-A$^b$ tetramers (NIH Tetramer Core Facility), VP1-221-235 and LT678-690, at 8 μg/mL of each tetramer or I-A$^b$ hCLIP at 16 μg/mL as a control. Cells were stained with the tetramers for 1.5 h at 37°C. Dead cells were identified by staining with Fixable Viability Dye (eBioscience). MHC-I tetramer staining of CD8$^+$ T cells was performed with APC-conjugated D$^b$-LT359 tetramer (NIH Tetramer Core Facility) at 4 μg/mL. Cells were stained with antibodies for the following markers: CD8α (53-6.7), CD4 (RM4-5), CD44 (IM7), CD45.1 (A20), CD45.2 (104), CCR4 (2G12), CCR5 (HM-CCR5), CCR6 (29-2L17), CCR7 (4B12), CCR9 (9B1), CXCR3 (CXCR3-173), CXCR4 (L276F12), CXCR5 (2G8), CXCR6 (SA051D1), CD69 (H1.2F3), Ki67 (SolA15). For intracellular/intranuclear staining, cells were fixed and permeabilized with a Fixation/Permeabilization kit (eBioscience). For Annexin V staining, cells were incubated with 0.6 μg/mL Annexin V (Biolegend). FLICA staining for Caspase-3/7 activity was performed according to the manufacturer's instructions (Bio-Rad). Flow cytometry samples were acquired on a BD FACSymphony A3 (BD Biosciences) and analyzed with FlowJo software (TreeStar).

## Quantification of viral DNA and qPCR

DNA was isolated from tissues using the Wizard Genomic DNA Purification Kit (Promega) and viral genome levels were quantified by TaqMan qPCR and compared to a standard curve [54]. RNA was isolated from tissues using TRIzol Reagent (ThermoFisher) and CXCL16 levels were quantified by TaqMan qPCR and normalized to TATA-box binding protein (IDT).

## Western blotting

Whole-cell kidney lysates were prepared with RIPA Lysis Buffer System supplemented with protease inhibitors (Santa Cruz Biotechnology). Protein concentrations were quantified by the Pierce BCA Protein Assay Kit (ThermoFisher Scientific). 20 μg of protein was boiled with SDS loading dye containing 2-betamercaptoethanol and run on 10% SDS-PAGE gels. Following transfer, polyvinylidene difluoride membranes were blocked in 5% blocking buffer prepared using non-fat dry milk dissolved in Tris-buffered saline (BioRad) supplemented with 0.1% Tween-20 (TBS-T). Membranes were incubated overnight at 4°C with rabbit anti-CXCL16 (Bioss) or rabbit anti-β-actin (Cell Signaling Technology) in 1% milk in TBS-T. Blots were washed with TBS-T followed by a 1.5 h incubation with horseradish peroxidase conjugated secondary antibodies (BioLegend) in 1% milk in TBS-T. After another wash with TBS-T, blots were developed with SuperSignal West Pico PLUS Chemiluminescence Substrate

(ThermoFisher Scientific). Western blot protein band intensity quantifications for sCXCL16 were normalized to the β-actin loading control and determined by ImageLab software (BioRad).

## ELISA

For measuring MuPyV-specific serum IgG levels, $1x10^6$ PFU of purified MuPyV was used as a capture antigen and incubated with serum. Bound IgG was detected with an anti-mouse IgG secondary (Bethyl Laboratories). IgG concentration was calculated using a standard curve of the VP1 mAb 8A7H5 [19,55].

## Immunofluorescence and immunohistochemical staining

For mouse kidney immunofluorescence, kidneys were immersion-fixed in neutral buffered formalin overnight prior to processing and paraffin-embedding. Sections were stained with antibodies against NKCC2 (Abcam), CD8 (Invitrogen, clone 4SM15), CXCL16, (Bioss) and GFP (Invitrogen), and mounted with Prolong Gold Antifade Mountant with DAPI (ThermoFisher). A formalin-fixed, paraffin-embedded sections of a kidney needle biopsy from a deidentified renal allograft patient was immunohistochemically stained with an anti-SV40 T antigen (clone MRQ-4, Cell Marque). For immunofluorescence staining, sections were stained with antibodies against CD8 (Abcam) and CXCR6 (Abcam).

## Data acquisition and processing

We identified studies in the Gene Expression Omnibus (GEO) database containing kidney biopsy samples from patients who had undergone kidney transplantation and subsequently developed BK virus nephropathy (cases), as well as samples from patients with stable graft function (controls). A total of 4 studies were identified, of which one was RNA-seq based and three were microarray based. The number of cases and controls varied from 3-15 and 5-68 respectively (Table 1). For the three microarray studies (GSE47199, GSE72925, and GSE75693), we downloaded the gene count matrix using the *getGEO* function from the GEO-query package v2.66.0 in R. We normalized the data using log-2 transformation. We ensured the data were normalized appropriately by ensuring the expression values were median centered across samples within a study. We then performed differential expression analysis between cases and controls using the limma package v3.54.2 in R. The differential expression p-values were adjusted for false discover rate (FDR). For the RNA-seq study (GSE120495), we downloaded the gene count matrix from GEO. We removed genes with <10 read counts in at least 5 samples. Read counts were normalized to transcripts per million (TPM). We then performed differential expression analysis between cases and controls using the DESeq2 package v1.38.3. The differential expression p-values were adjusted for false discover rate (FDR). To integrate the results from multiple studies, we performed a meta-analysis using Cauchy combination1 method with uniform weights to integrate the p-values across these studies. This method combined the p-values from each study to identify genes with consistent differential expression across multiple studies.

## Statistical analysis

All data are displayed as mean ± standard deviation. The statistical tests performed are listed in the respective Figure legends and were carried out using Prism (Graphpad). Exact p values and sample sizes are listed in the Figures and values < 0.05 were considered significant. Samples sizes were not statistically pre-determined and all samples represent individual mice or other biological replicates.

## Supporting information

**S1 Fig. Phenotypic characterization of vascular and kidney-infiltrating CD8$^+$ and CD4$^+$ T cells.** (A) Expression of CD69 on intravascular and infiltrating CD8$^+$ CD44$^+$ D$^b$-LT359 tetramer$^+$ T cells 30 dpi. Data are representative of three independent experiments (n = 5). (B) Expression of CD69 on intravascular and infiltrating CD4$^+$ CD44$^+$ T cells 30 dpi. Data are representative of three independent experiments (n = 5). Data are from two independent experiments (n = 10). Data were analyzed by paired $t$ tests.
(TIF)

**S2 Fig. Lack of CXCR6 disrupts kidney T cell responses and virus control.** (A) Comparison of CXCR6 expression on intravascular and infiltrating kidney CD8$^+$ and CD4$^+$ WT and CXCR6$^{-/-}$ T cells. (B) Number of CD8$^+$ CD44$^+$ D$^b$-LT359 tetramer$^+$ T cells in the spleens of WT and CXCR6$^{-/-}$ mice 8, 15, and 30 dpi. Data are from 4-5 independent experiments (n = 16-21). (C) Number of CD4$^+$ CD44$^+$ T cells in the spleens of WT and CXCR6$^{-/-}$ mice 8, 15, and 30 dpi. Data are from 4-5 independent experiments (n = 16-21). (D) Virus levels in the spleens of WT and CXCR6$^{-/-}$ mice 8, 15, and 30 dpi. Data are from four independent experiments (n = 16-19). (E) Virus levels in the kidney of WT and CXCR6$^{-/-}$ mice 4 dpi. Data are from three independent experiments (n = 12-13). (F) Virus levels in the spleen of WT and CXCR6$^{-/-}$ mice 4 dpi. Data are from three independent experiments (n = 12-13). (G) Virus levels in the liver, lung, and salivary gland of WT and CXCR6$^{-/-}$ mice 8, 15, and 30 dpi. Data are from 3-4 independent experiments (n = 12-20). (H) ELISA for anti-capsid IgG in the serum of WT and CXCR6$^{-/-}$ mice 8, 15, and 30 dpi. Data are from 2-3 independent experiments (n = 8-13). Data were analyzed by multiple Mann-Whitney tests (B-D, G), Mann-Whitney tests (E-F), and multiple $t$ test (H).
(TIF)

**S3 Fig. Proliferation and apoptosis of kidney-infiltrating T cells.** (A) Frequency of Ki67$^+$ WT and CXCR6$^{-/-}$ TCR-I cells in the spleen of recipients 8, 15, and 30 dpi. Data are from two independent experiments (n = 10). (B) Kidney-infiltrating CD8$^+$ CD44$^+$ D$^b$-LT359 tetramer$^+$ and CD4$^+$ CD44$^+$ T cells stained with FLICA to identify activated caspase 3/7. Data are from two independent experiments (n = 6-9). (C) Kidney CD8$^+$ and CD4$^+$ T cells stained with viability dye and Annexin V to identify apoptotic (Annexin V$^+$ viability dye$^-$) cells in WT and CXCR6$^{-/-}$ mice 8 dpi. Data are from two independent experiments (n = 7-9). (D) Frequency of CD8$^+$ and CD4$^+$ cells out of total CD45$^+$ cells in the blood of WT and CXCR6$^{-/-}$ mice treated with vehicle or FTY720. Data are from three independent experiments (n = 10-13). (E) Frequency of splenic CD8$^+$ CD44$^+$ D$^b$-LT359 tetramer$^+$ and CD4$^+$ CD44$^+$ T cells that are Ki67$^+$ in WT and CXCR6$^{-/-}$ mice with vehicle or FTY720 treatment. Data are from three independent experiments (n = 10-13). Data were analyzed by multiple paired t tests (A), unpaired $t$ tests (B-C), or multiple unpaired $t$ tests (D-E).
(TIF)

**S4 Fig. IL-12 induces CXCR6 on T cells.** (A) Frequency of CXCR6 expression on D$^b$-LT359 tetramer$^+$ and D$^b$-LT359 tetramer$^-$ CD8$^+$ CD44$^+$ T cells in the spleens of WT mice 8, 15, and 30 dpi. Data are from two independent experiments (n = 7). (B) CXCR6 expression on spleen CD8$^+$CD44$^+$ D$^b$-LT359 tetramer$^+$ T cells and CD4$^+$CD44$^+$ T cells after treatment with PBS vehicle or IL-12. Data are from three independent experiments (n = 14-15). (C) CD8$^+$ CD44$^+$ D$^b$-LT359 tetramer$^+$ T cell and CD4$^+$ CD44$^+$ T cell numbers after treatment with vehicle or IL-12. Data are from two independent experiments (n = 9). (D) Virus DNA genomes in spleen after vehicle or IL-12 treatment. Data are from three independent experiments (n = 14-15). Data were analyzed by two-way ANOVA (A) or unpaired $t$ test (B-D).
(TIF)

## Acknowledgements

We thank the staff of the Flow Cytometry facility (RRID:SCR_021134), the Comparative Medicine Histology Core, and the Department of Comparative Medicine.

## Author contributions

**Conceptualization:** Matthew D. Lauver, Aron Lukacher.

**Funding acquisition:** Havell Markus, Dajiang J Liu, Aron Lukacher.

**Investigation:** Matthew D. Lauver, Zoe E Katz, Havell Markus, Nicole M Derosia, Ge Jin, Katelyn N Ayers, Arrienne B Butic.

**Methodology:** Matthew D. Lauver, Zoe E Katz, Havell Markus.

**Project administration:** Matthew D. Lauver, Zoe E Katz, Aron Lukacher.

**Resources:** Kaitlyn Bushey.

**Supervision:** Dajiang J Liu, Aron Lukacher.

**Visualization:** Matthew D. Lauver, Catherine S Abendroth.

**Writing – original draft:** Matthew D. Lauver, Zoe E Katz.

**Writing – review & editing:** Zoe E Katz, Aron Lukacher.

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
