## [Decision Letter · Decision Letter 0]

22 Jan 2025

PPATHOGENS-D-24-02707

The CXCR6-CXCL16 axis mediates T cell control of polyomavirus infection in the kidney

PLOS Pathogens

Dear Dr. Lukacher,

Thank you for submitting your manuscript to PLOS Pathogens. After careful consideration, we feel that it has merit but does not fully meet PLOS Pathogens's publication criteria as it currently stands. Therefore, we invite you to submit a revised version of the manuscript that addresses the points raised during the review process.

Please submit your revised manuscript within 30 days Mar 23 2025 11:59PM. If you will need more time than this to complete your revisions, please reply to this message or contact the journal office at plospathogens@plos.org. Please include the following items when submitting your revised manuscript:

We look forward to receiving your revised manuscript.

Kind regards,

Christopher M. Snyder, Ph.D.

Guest Editor

PLOS Pathogens

Blossom Damania

Section Editor

PLOS Pathogens

Sumita Bhaduri-McIntosh

Editor-in-Chief

PLOS Pathogens

orcid.org/0000-0003-2946-9497

Michael Malim

Editor-in-Chief

PLOS Pathogens

orcid.org/0000-0002-7699-2064

**Additional Editor Comments:**

Dear Dr. Lukacher. Please find the reviews below. All three reviewers found the article to be significant and well-designed. However, they each raised multiple points for clarification of the data and interpretations, and/or suggestions for improving the figures. Please address these points in your revised manuscript.

**Journal Requirements:**

At this stage, the following Authors/Authors require contributions: Matthew Lauver, Zoe Katz, Havell Markus, Nicole Derosia, Ge Jin, Katelyn Ayers, Arrienne Butic, Kaitlyn Bushey, Catherine Abendroth, Dajiang Liu, and Aron Lukacher. Please ensure that the full contributions of each author are acknowledged in the "Add/Edit/Remove Authors" section of our submission form.

3) Please ensure that the funders and grant numbers match between the Financial Disclosure field and the Funding Information tab in your submission form. Note that the funders must be provided in the same order in both places as well. State the initials, alongside each funding source, of each author to receive each grant. For example: "This work was supported by the National Institutes of Health (####### to AM; ###### to CJ) and the National Science Foundation (###### to AM).".

**Reviewers' Comments:**

Reviewer's Responses to Questions

**Part I - Summary**

Reviewer #1: In this clearly written manuscript, Lauver et al. present evidenceCxCR6/CxCL16 and IL-12 help drive T cell homing to the kidney during infection with polyomaviruses. There are several strengths to this work that include addressing a clinically important problem, results that suggest future clinical interventions, demonstrating plausibility using human data of careful mechanistic studies in the murine model, and having several important findings being supported by orthogonal approaches. The numerous mechanistic details addressed in this work give confidence to the overall conclusions. Critiques include mostly suggestions for adding wording in the name of transparency and for clarity on a couple of figures.

Critiques/Suggestions

1. Can the authors add a sentence or two to the discussion addressing possible model for why the mechanism they have discovered for muPyV and the similar reported mechanism for LCMV appear specific to the kidney. This is a fascinating observation. Why the kidney? Is the kidney somehow “special” compared to others organs/tissues. If not, how many other organs/tissues have a specific chemokine code for homing T-cells?

2. Line 107, for clarity, consider explaining the rationale for the composition of the panel of “variety” of chemokine receptors that were included. As worded and from the data shown in Fig. 1A, it is difficult to discern whether those that scored were “cherry picked” from many that could have scored.

3. Line 134, Fig. 2C., my image may be poor, but to my eye the purported increase 35kD fragment of CxCL16 is not obvious in all 5 mice – perhaps this is only clear in 3 of the 5 mice and the magnitude is such that it could be within the margin of lane loading differences. Can the authors rule out loading variation? Adding text to the manuscript addressing this would be appropriate as I suspect other readers might have a similar impression. Normalized quantification would be required to fully address this critique. Minimally, a clearer image could be included.

Reviewer #2: Lukacher and colleagues provide a manuscript demonstrating the importance of CXCL16 in recruiting T cells into kidneys of mice infected with the mouse polyoma virus PyV as a model for humans infected with BK polyoma virus. The authors provide convincing evidence that CXCL16 expression in the kidney’s aids in recruiting CD4 and CD8 T cells into the kidneys of infected mice as well as retaining CD8 T cells and this is associated with control of viral replication. The authors provide evidence that IL-12 administration increases CXCR6 on virus-specific CD8+ T cells enhancing accumulation within the kidneys of infected mice. The authors present human data arguing for a role for CXCR6 on CD8+ T cells within the kidney biopsies of patients with PyV-associated nephropathy. Overall, this a well-conceived study from an excellent laboratory with a long-standing history of innovative work in the field of polyoma virology/pathogenesis. While the study provides an important step forward in understanding host defense and potential mechanisms contributing to kidney disease, there are several items that require clarification. Two key points the authors should address is i) the human data is not convincing as presented in terms of the role of the CXCL16:CCR6 signaling axis in CD8+ T cell recruitment to the kidneys and ii) if the authors could discuss in more detail how targeting this chemokine signaling pathway could be used for ameliorating disease in humans it would increase the potential clinical importance of the findings provided.

Reviewer #3: The manuscript by Lauver et. al. investigates the factors that regulate T cell migration and retention in the kidneys after polyomavirus infection. The authors show that the chemokine receptor CXCR6 is highly expressed on infiltrating virus-specific CD4 and CD8 T cells within the kidney, and CXCR6 deficiency or blockade of CXCL16 inhibits the recruitment or retention of CD4 and CD8 T cells in the kidney, respectively. They also demonstrate the IL-12 administration enhances CXCR6 expression, resulting in decreased viral load. Finally, analysis of existing datasets from kidney transplant patients showed increased expression of both CXCR6 and CXCL16.

The overall conclusions of the manuscript are well supported by the data, and the work addresses an important question in the field. Investigating the CXCR6-CXCL16 axis from multiple angles is a strength throughout the manuscript and tying the findings from the mouse model to PVAN in humans adds clinical relevance. I have no major concerns related to scholarship or interpretations of experiments, but several minor issues that can be addressed from the existing data.

**Part II – Major Issues: Key Experiments Required for Acceptance**

Reviewer #1: (No Response)

Reviewer #2: No significant major issues i.e. additional experiments required from my point of view

Reviewer #3: None

**Part III – Minor Issues: Editorial and Data Presentation Modifications**

Reviewer #1: (No Response)

Reviewer #2: Two key points the authors should address is i) the human data is not convincing as presented in terms of the role of the CXCL16:CCR6 signaling axis in CD8+ T cell recruitment to the kidneys and ii) if the authors could discuss in more detail how targeting this chemokine signaling pathway could be used for ameliorating disease in humans it would increase the potential clinical importance of the findings provided.

Reviewer #3: 1. The class II tetramer data demonstrating CXCR6 expression on virus-specific CD4 T cells in the kidney is important and should be moved from the supplement figures to figure 1.

2. In figure 3e, are the comparisons between WT and CXCR6 KO cells made comparing total TCR-I T cells, or only cells with a TRM phenotype (CD69+ CD103+)? If made with total T cells, is there a similar defect observed in CXCR6 KO cells when looking at only cells with a TRM phenotype?

3. The graph in figure 6d is difficult to interpret. Are all these data points significant, and what are the comparisons being made? The summary in supplemental table 1 is more clear, could these data be included in figure 6?

PLOS authors have the option to publish the peer review history of their article (what does this mean? ). If published, this will include your full peer review and any attached files.

**Do you want your identity to be public for this peer review?** For information about this choice, including consent withdrawal, please see our Privacy Policy .

Reviewer #1: No

Reviewer #2: No

Reviewer #3: No

**Figure resubmission:**
---

## [Editor Report · Decision Letter 1]

10 Feb 2025

Dear Dr. Lukacher,

We are pleased to inform you that your manuscript 'The CXCR6-CXCL16 axis mediates T cell control of polyomavirus infection in the kidney' has been provisionally accepted for publication in PLOS Pathogens.

Best regards,

Christopher M. Snyder, Ph.D.

Guest Academic Editor

PLOS Pathogens

Blossom Damania

Section Editor

PLOS Pathogens

Sumita Bhaduri-McIntosh

Editor-in-Chief

PLOS Pathogens

orcid.org/0000-0003-2946-9497

Michael Malim

Editor-in-Chief

PLOS Pathogens

orcid.org/0000-0002-7699-2064

The authors have addressed the reviews, which were all minor, and improved the clarity of the manuscript.
---

## [Editor Report · Acceptance letter]

Dear Dr. Lukacher,

We are delighted to inform you that your manuscript, "The CXCR6-CXCL16 axis mediates T cell control of polyomavirus infection in the kidney," has been formally accepted for publication in PLOS Pathogens.

Best regards,

Sumita Bhaduri-McIntosh

Editor-in-Chief

PLOS Pathogens

orcid.org/0000-0003-2946-9497

Michael Malim

Editor-in-Chief

PLOS Pathogens

orcid.org/0000-0002-7699-2064